# Differential expression of gut protein genes and population density of *Arsenophonus* contributes to sex-biased transmission of *Bemisia tabaci* vectored *Cotton leaf curl virus*

Ikbalpreet Singh[1], Ramandeep Kaur[2], Ashok Kumar[2], Satnam Singh [2]*, Abhishek Sharma[3]

1 Department of Plant Pathology, Punjab Agricultural University, Ludhiana, India, 2 Punjab Agricultural University, Regional Research Station, Faridkot, Punjab, India, 3 Department of Vegetable Sciences, Punjab Agricultural University, Ludhiana, India

☯ These authors contributed equally to this work.
* satnam@pau.edu

**Data Availability Statement:** All relevant data are within the manuscript and its Supporting information files.

## Abstract

Whitefly, *Bemisia tabaci* (Gennadius) is an important pest of cotton causing direct damage as sap feeder and vector of *Cotton leaf curl virus* (CLCuV). Previous few studies suggest that female whiteflies are more efficient vector of begomovirusthan males, however the sex-biased transmission efficiency is still not clearly understood. Present studies with *B. tabaci* AsiaII-1 haplotype showed higher virus transmission efficiency of females compared to males. This variable begomovirus transmission efficiency has been related to previously identifiedkey factors associated with *B. tabaci*. The higher density of endosymbiont *Arsenophonus* and variable expression of some midgut proteins genes i.e. *Cyclophilin*, *Knottin*, *Hsp40*, *Hsp70* may be possibly imparting higher vector competency to the females compared to males. The present studies suggest low abundance of *Arsenophonus* spp. as well as lower expressionof *Cyclophilin* genein males as compared to females. This is further supplemented by overexpression of *Knottin*, *Hsp40*, and *Hsp70* genes in males compared to females and thus collectively all these factors might be playing a key role in low virus transmission efficiency of males. The relative density of *Arsenophonus* spp. and expression of midgut proteins genes in male and female whitefly first time enriches our understanding about sex-biased transmission efficiency of begomovirus.

## Introduction

The cotton whitefly, *Bemisia tabaci* (Gennadius) (Hemiptera: Aleyrodidae) comprises a complex of morphologically indistinguishable species that cause considerable losses to numerous crops both as pest and vector of plant viruses globally [1]. Whitefly causes damage to

**Funding:** The author(s) received no specific funding for this work.

**Competing interests:** The authors have declared that no competing interests exist.

agricultural crops by direct sap-feeding, secreting honeydew which leads to saprophytic growth on the host plant that interferes with the photosynthetic ability of plants and as a vector of begomoviruses. Higher fecundity, more dispersal, polyphagy, development of insecticide resistance are the driving factors that have made this insect a global pest [2–4]. Whitefly infests more than 500 host plants which include fiber crops, ornamentals, vegetables, legumes, and weeds and vectors 114 plant viruses belonging to genera begomoviruses (*Geminiviridae*), criniviruses (*Closteroviridae*), torrado viruses (*Secoviridae*), and ipomoviruses (*Potyviridae*) [5]. Begomoviruses constitute the most damaging viruses which have emerged in recent decades to affect most of the fiber, ornamental and vegetable crops globally [5]. Members of the genus Begomoviruses belong to the family geminiviridae and consist of either ssDNA or dsDNA, which are exclusively transmitted by whiteflies in a circulative persistent manner [6]. Begomoviruses and vector whiteflies have co-evolved as a complex interacting system; it is still not completely understood that persistent transmission of the virus is propagative (virus replicates inside host-vector) or non-propagative (virus do not replicate inside host-vector). Persistent circulative passage of Begomoviruses inside the vector involved complex interaction of viral coat protein with host vector proteins [7–9]. Among the persistently transmitted viruses *Cotton leaf curl virus* (CLCuV) species are highly devastating and poses a serious threat to the fiber sector by causing Cotton leaf curl virus disease (CLCuD). CLCuD is a serious problem in African cotton growing regions, north-Indian cotton-growing states, Pakistan, Bangladesh, Philippines and China [10]. Recently CLCuV has been classified into five major species which include *Cotton leaf curl Kokharan virus* (CLCuKoV), *Cotton leaf curl Multan virus* (CLCuMuV), *Cotton leaf curl Allahabad virus* (CLCuAlV); *Cotton leaf curl* Bangalore virus (CLCuBaV) and *Cotton leaf curl Gezira virus* (CLCuGeV) present across different cotton growing regions of world [11]. Both virus and vector have strong molecular interactions and due to a wide host range of species complex new viral species are evolving, which pose a serious threat to cotton cultivation in many cotton growing countries. Many earlier reports have compared the differential transmission of Begomoviruses by different haplotypes of *B. tabaci* species complex. The vector competence of the Asia II-1 haplotype, a predominant in north India for transmitting CLCuD is higher when compared with other haplotypes [12]. The Asia II-1 haplotype has been found to be closely associated with CLCuD incidence in the north-western region of India, while south India native population belongs to Asia-1 and recorded CLCuD incidents are very low when compared to the northwestern zone. This disease-specific geographic distribution pattern of whitefly haplotypes had prompted several questions regarding the transmission of CLCuV. Previous studies revealed the interaction of GroEL protein secreted by secondary symbionts *Hamiltonella* and *Arsenophonus* of whiteflies with viral coat protein [13]. Interrupting the functioning of GroEL had resulted in decreased virus transmission. Similarly reports have been published suggesting the role of several genes in circulative virus transmission mediated by whitefly [14–16]. It has been evidenced that females are efficient vectors in comparison to male based on variable symbiont density [17]. Even under different crops (tomato, okra, cotton, cucurbits, and chili) germplasm screening against begomovirus, the use of female *B. tabaci* as a vector has been documented [18]. This led us to investigate the question whether CLCuV transmitted from Asia II-1 whiteflies sex-dependent manner. However, to date only a couple of studies have been taken up on focusing on sex-biased differential transmission by whitefly. The present study reports the differential virus transmission with males and females of Asia II-1 haplotype of *B. tabaci* in context to endosymbiont *Arsenophonus* spp. and genes associated with virus transmission in whitefly. The finding of this study will advance our understanding regarding the sex-biased key players involved in CLCuV transmission.

## Materials and methods

### Raising of whitefly and virus cultures

The susceptible variety of cotton *Gossypiumhirsutum* cultivar RST9 was sown in pots in insect-proof cages for growing virus-free plants (S1 Fig). Staggered sowing of cotton was done for a continuous supply of plants for the begomovirus transmission study and whitefly culture maintenance. Non-viruliferous whitefly haplotype Asia II-1 was continuously maintained on these plants at 26 ± 1 ˚C, 60%RH, and 14 h light/10 h darkness under insect-proof cages (Bug-Dorm- MegaView Science Co., Ltd., Taiwan). The virus culture of most prevalent *Cotton leaf curl Multan virus*-Rajasthan (CLCuMuV-Ra) (Accession no. MZ365008) was also maintained in these plants kept in insect-proof cages.

### Quantification of viral load in whitefly males and females

Non-viruliferous whitefly adults (1–2 days old) were collected in a 50 ml falcon tube and subjected to 2 h starvation and then given the feeding acquisition of 48 h on virus-infected plant leaves in closed cups covered with fine muslin cloth on one side (S2 Fig). After 48 h the live whiteflies were collected with the help of an aspirator wrapped outside with aluminum foil and kept in -20˚C for 5 min to reduce their activity and transferred to a 50 ml falcon. The tube was placed continuously on ice and the flies were sexed under a stereo zoom microscope (Olympus SZX7) in small batches with the help of a '0' size sable hairbrush. Morphological distinction was done based on size and shape of ovipositor (S3 Fig). The males and females were transferred into separate 1.5 ml tubes with 250 μl TRI Reagent® RNA Isolation Reagent (Sigma-Aldrich, Inc.) and proceeded for RNA isolation. Each sample consisted of five biological and three technical replicates. The integrity of the RNA was determined by Biospectrometer (Eppendorf). Total RNA (1μg) was reverse transcribed into cDNA using Primescript First-strand cDNA synthesis kit (Takara). The cDNA was diluted 10 times prior to reaction setup and each qPCR reaction consisted of 1 μl cDNA, 0.1 μL of each primer (10 μM), and 5 μl of SYBR® Premix *Ex Taq*TM II in a total volume of 10 μl.

The differential retention of CLCuMuV-Ra by female and male whitefly was assessed using quantitative real-time PCR (qRT-PCR) by quantifying viral coat protein (CP) copies on a Light cycler System (Roche Life Sciences, Mannheim, Germany) using SYBR® Premix Ex TaqTMII (Takara) according to manufacturer's protocol. The reaction conditions were 30 s at 94˚C followed by 40 cycles consisting of 30 s at 95˚C, 5s at 95˚C, and 30 s at 60˚C. The accompanying software LightCycler® 96 Application Software Version 1.1 was used for qPCR data normalization and quantification. The relative virus titer was inferred on the basis of relative expression level of the viral CP gene calculated using the $2^{-\Delta\Delta Ct}$ method. The expression level of different genes also calculated using ΔCT method and presented in terms of means± SEM. The expression data was analyzed using Student t- test at $p < 0.05$. All the gene expression data in this study was normalized using β-*tubulin* gene of *B. tabaci*as housekeepinggene based on our previous studies [19]. To rule out the non-specific binding/ background amplification, qRT-PCR reactions were setup using non-viruleferous and viruleferous whitefly, as well as CLCuMuV-Ra infected and non-infected cotton plants. The RT-PCR primers for the coat protein were designed using Primer 3 Software using viral sequence (Acc. # MZ36500) The details of the primers used in the study are given in Table 1.

### Relative feeding and fluid loss assay

Fifty males and females were allowed to feed on 100 μl sucrose diet in three different replicates. For this a 50 ml falcon tube open at both ends was used as a feeding chamber. It was covered

**Table 1. Details of primers used in the study.**

| Target gene | Primer sequence 5'-3' | Expected size (bp) |
|---|---|---|
| q_Cycloph_F | GACGTAGGTCAAGATCCAGAGA | 133 |
| q_Cycloph _R | GAGGAAACTGCTCGTCCTTT | |
| q_HSP40_F | CTGTAGAAAGGATCCC | 145 |
| q_HSP40_R | AACACCGTTGCGACTTACAA | |
| q_HSP70_F | AGTGCGGACGAACTAGCACT | 117 |
| q_HSP70_R | GCAGCCAAATGATCAAGTCA | |
| q_Knottin_F | GGCAATTGTCTTCCTGACG | 120 |
| q_Knottin_R | TGTGCAATGTCCAGGAGTTC | |
| qCLCuV_F | CGTCGACCTGTTGATAAACCTC | 150 |
| qCLCuV_R | GCATATTGACCA CCGGTAACAG | |
| qArs_F | GGTGACAGCCCCGTATCTAA | 181 |
| qArs_R | TTCCCTCACGGTACTGGTTC | |
| β-*tubulin*_F | CACTGGTACGTAGGAGAAGGTA | 117 |
| β-*tubulin*_R | ACTGAGTCCATGCCAACTTC | |
| ds_Hsp40_F | TAATACGACTCACTATAGGGCTCAAGCGTACGAAGTCCTATC | 371 |
| ds_Hsp40_R | TAATACGACTCACTATAGGGGTGGTGATGATGGTGGTTACT | |
| ds_Hsp70_F | TAATACGACTCACTATAGGGCTCGAAACAAGCGAGGAGATT | 373 |
| ds_Hsp70_R | TAATACGACTCACTATAGGGCGATGGAGAGCAGCATCAATT | |
| ds _Cycloph_F | TAATACGACTCACTATAGGGCAAGAGAGACAATCCAG | 468 |
| ds _Cycloph_F | TAATACGACTCACTATAGGGCCTGAGTGGAACATATC | |
| ds_Knottin_F | TAATACGACTCACTATAGGACGACCAAAGCTTATC | 500 |
| ds_Knottin _R | TAATACGACTCACTATAGGATCTGGTTGTGCAATG | |
| qGroEL_F | TCGGTACGATTTCGGCTAAC | 170 |
| qGroEL_R | AATAACCGCGATCAAATTGC | |

by muslin cloth at one end and with stretched parafilm at the other end. Sucrose diet solution (20%) was placed on the parafilm and covered with another layer to enclose the diet solution (S3 Fig). The feeding chamber without whiteflies was considered as control and used to estimate the diet loss in 24 h due to evaporation. The amount of diet left in all the treatments and control was quantified using Nanoject (Durmmond Scientific Inc, USA). The diet loss in control was subtracted from both the treatment and total diet consumed per whitefly adult was calculated in nl. Similarly for estimating the amount of fluid excretion from 25 male and female whitefly adults in three replicates from above experiment were released on ventilated Petri plate with water sensitive paper (TeeJet®Technologies) 50mm in diameter placed on inner side covering the whole base. After 1 h the number of blue dots were calculated on each paper disc under Stereo Zoom Microscope (SZX7, Olympus) to estimate the comparative fluid loss.

## Comparative transmission and quantification of viral load in host plant

Non-viruliferous AsiaII-1 whiteflies were starved for 2 h followed by feeding acquisition of 48h on virus-infected plants. Adults were sexed as per methodology described in the previous section and 10 males and 10 females were used separately to inoculate freshly raised virus-free cotton plants at 3 or 4 true-leaf stage (3 weeks after sowing) for 48h in leaf clip cages (S1 Fig). The study was conducted with four biological replicates with three plants in each. Post-acquisition after 48 h the whiteflies were removed and further, the plants were kept whitefly free by spraying Diafenthiuron 50% WP (Syngenta India Ltd.) thoroughly on each of the virus inoculated cotton plants. The plants were kept in insect-proof cages for observation of symptoms. Total

RNA was extracted using TRI Reagent® RNA Isolation Reagent (Sigma-Aldrich, Inc.) from the male and female inoculated plants after the appearance of the first viral symptoms. The quality and quantity of RNA was checked through a spectrophotometer (Eppendorf BioSpectrometer® basic). The RNA was reverse transcribed into cDNA using PrimeScript 1st strand cDNA Synthesis Kit (Takara). The cDNA samples were diluted tenfold prior to reaction setup and the relative expression of the virus was studied using viral CP gene-specific primers. The qPCR reaction was carried out in Light cycler System (Roche life sciences, Mannheim, German) SYBR® Premix Ex TaqTMII, (Takara) according to manufacturer's protocol with thermal cycling conditions which constitute of initial denaturation of 95 ˚C for the 30s followed by 40 cycles of 95 ˚C for 5s and 60 ˚C for 10s. The expression of the target gene was normalized using the *histone 3* gene of cotton [20] Statistical analysis was performed by using the student's t-test.

## Relative expression of *Hsp70, Hsp 40, Knottin* and *Cyclophilin* genes and density of *Arsenophonus* spp. in male and females whiteflies and its impact viral load

The relative expression of *Hsp70*, *Hsp40*, *Knottin* and *Cyclophilin* was estimated in both viruliferous and non-viruliferous male and female whiteflies. For this 2–3 days old 10 whiteflies each was subjected to total RNA isolation using TriReagent® (Sigma-Aldrich) as per manufacturer's protocol followed by DNAse treatment. The quality and quantity of RNA were checked on the Eppendorf BioSpectrometer® basic. First-strand cDNA was prepared from 1 µg of RNA using the PrimeScript™ 1st strand cDNA Synthesis Kit (Clontech Takara) as per protocol. The expression of *Hsp70*, *Hsp40*, *Knottin*, and *Cyclophilin* genes was quantified using SYBR-Green (SYBR® Premix Ex TaqTMII, Takara) and respective gene specific primers (Primer3 software) (Table 1) in qRT-PCR LightCycler® 96 System (Roche life sciences, Mannheim, Germany). The cDNA was diluted 10 times prior to reaction setup and each qPCR reaction consisted of 1 µl cDNA, 0.1 µL of each primer (10 µM), and 5 µl of SYBR® Premix *Ex Taq*TM II in a total volume of 10 µl. The reaction conditions were 30 s at 94˚C followed by 40 cycles consisting of 10 s at 94˚C, 30 s at 55˚C, and 20 s at 72˚C. Similarly, the density of *Arsenophonus* in both viruliferous and non-viruliferous males and females (sampled as per earlier mentioned protocol in this section) was quantified through the relative expression of the *23SrRNA* gene of the bacterium using qRT-PCR based quantification. Each biological sample consisted of five replicates. All these qRT-PCR reactions were normalized using constitutively expressed *B. tabaci -β tubulin* as a housekeeping gene.

## Elimination of *Arsenophonus* and dsRNA-mediated silencing of *Hsp70, Hsp 40, Knottin,* and *Cyclophilin* genes to study their impact on viral load in whitefly

The endosymbiont *Arsenophonus* spp. was eliminated using tetracycline feeding to adult whiteflies through an artificial diet-feeding approach. A 50 ml falcon tube open at both ends was used as a feeding chamber covered by muslin cloth at one end and with stretched parafilm at the other end. Sucrose diet solution (20%) infused with tetracycline (90 µg/ml) was placed on the parafilm and covered with another layer to enclose the diet solution (S3 Fig). The selection of antibiotic and its dose was inferred from our previous study which showed complete elimination of *Arsenophonus* from whitefly after feeding of tetracycline-infused sucrose diet for 48 h. Post 48h feeding, male and female were separated as described in earlier sections. *Arsenophonus* cured ten individual female and male adults were subjected to qRT-PCR detection as described in the previous section to confirm viral load in each group with each biological sample replicated five times.

In order to study the impact of selected genes on the viral load in whitefly template for dsRNA against respective genes was amplified from cDNA using gene-specific primers having T7 promoter sequence (5′–TAATACGACTCACTATAGGG–3′) at the 5′ ends of both reverse and forward primer. Amplicons were gel purified using NucleoSpin Gel and PCR Clean-up Kit (Macherey-Nagel) as per the manufacturer's protocol. The respective amplicons were used as a template for *in vitro* transcription using TranscriptAid T7 High Yield Transcription Kit (Fermentas) as per the kit manual. The quality and quantity of dsRNA was determined by agarose gel electrophoresis and Eppendorf BioSpectrometer ® basic, respectively.

Viruliferous adult whiteflies sampled as per earlier described protocol were given feeding access to dsRNA (400 ng/ ul of diet) against *Hsp70*, *Hsp40*, *Knottin*, and *Cyclophilin*genes incorporated in 20% sucrose diet. The dose and feeding methodology was followed as per our earlier studies [21, 22]. A similar quantity of dsRNA against green fluorescent protein gene (dsGFP) was used as control. Post 48 h of feeding accesses to dsRNA-sucrose diet mixture, male and female were separated under a stereomicroscope microscope as per methodology described in earlier sections. Each individual sex comprising of ten male and ten female whiteflies representing one biological sample replicated five times were collected in TriReagent® (Sigma-Aldrich) and subjected to RNA extraction followed by qRT-PCR analysis as per earlier described methodology. The relative fold change in viral CP gene expression was quantified in both male and female whiteflies using the delta-CT method. The gene expression was normalized withβ-*tubulin* as a housekeeping gene and compared with dsGFP fed whiteflies as control. Statistical analysis was performed by using the student's t-test. For all the primers used in the study dissociation curve or melt curve analysis was performed to check the non-specific amplification as well as or primer dimer if any.

## Result and discussion

### Effect of whitefly sex on begomovirus transmission efficacy

The sex dependant differential variation in the retention and transmission capability of CLCu-MuV-Rawas quantified using quantitative real-time PCR through the relative expression of CLCuMuV-Rain *B. tabaci* and its titer in cotton plant post transmission by female and male. The primer specificity was clearly indicated by the no amplification observed with these primers both in non-infected cotton plants and non-viruleferous whitefly compared infected cotton plants and viruleferous whitefly (Fig 1A and 1B).

The differential transmission of Begomoviruses (CLCuV) was studied in relation to male and female whiteflies. No significant difference was observed in transmission efficiency mediated by the number of whiteflies. No significant difference was observed in CLCuMuV-Ra expression in both female and male whiteflies n = 5 and n = 20, as the single whitefly tend to retain the same amount of CLCuMuV-Ra titer but when we compare the expression of CLCu-MuV-Ra on sex basis, female tends to retain higher virus load (80%) compared to males (Fig 1C and 1D). Variations in male and female size may be related to the variable feeding capacity of each sex and this, in turn, may attribute to higher virus load acquired by the females. The feeding assay gave an indication of more feeding by females compared to males, however this was statistically at par with each other (Fig 1E). On an average the female and male whitefly consumed 68.06 nl and 48.73 nl of sucrose diet in 24 h, respectively. The average amount of diet lost due to evaporation in control was 7.1 μl from total of 100 μl (Fig 1E). Feeding was further validated indirectly through comparative amount of fluid excretion by males and females. The quantitative fluid loss by females was more as compared males, which is an indicative of more excretion by females possibly due to more feeding compared to males (Fig 1E). The average number of blue dots on female released water sensitive paper were 0.0097/ mm$^2$ compared

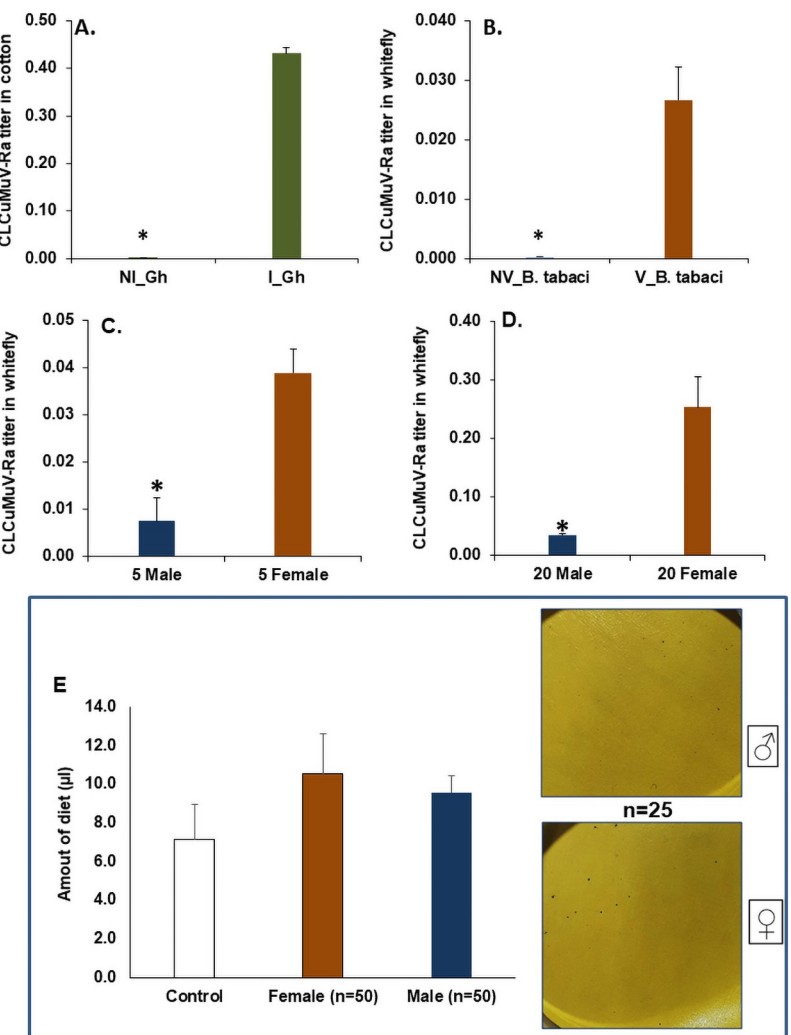

**Fig 1.** Validating the specificity of *Cotton leaf curl Multan virus*-Rajasthan primers in A. Non-infected (NI) and Infected *Gossypium hirusutum*; B. Non-viruliferous (NV) and Viruliferous (V) *Bemisia tabaci*; Relative amount of CLCuMuV-Ra in *B. tabaci*, C. n = 5 males and 5 females, D. n = 20 males and 20 females; E. Relative amount of sucrose diet fed by male and female whiteflies (n = 50), the control bar represents the diet loss due to evaporation. The fluid loss assay on water sensitive paper represented by number of blue dots shows the qualitative estimation of excretion by male and females (n = 25). The expression level (A-D) was normalized with *B. tabaci β-tubulin* as housekeeping gene. The error bars represent the standard deviation (n = 3) and *represents significant differences (P ≤ 0.05, Student's t-test).

to 0.0036 dots/ mm$^2$ in males. The intensity as well as the size of these dots was prominent in females compared to males. The current studies with feeding assays and fluid loss are indicative of more feeding and consequently more excretion in females compared to males.

Thus at the first instance difference in viral load or virus retention in *B. tabaci* is sex-biased with female whiteflies tend to retain statistically significant higher viral load compared to males. Followed by quantification of virus titer, the whitefly males and females released on the test plants indicated that in female fed plants vein thickening was apparent after 14 DPI (Days Post Inoculation), whereas at this time point the male fed plants show no visual symptom of CLCuMuV-Ra (Fig 2A). Furthermore, after 25 and 32 DPI downward curling and enation, respectively were observed in female inoculated test plants, however at the same time male

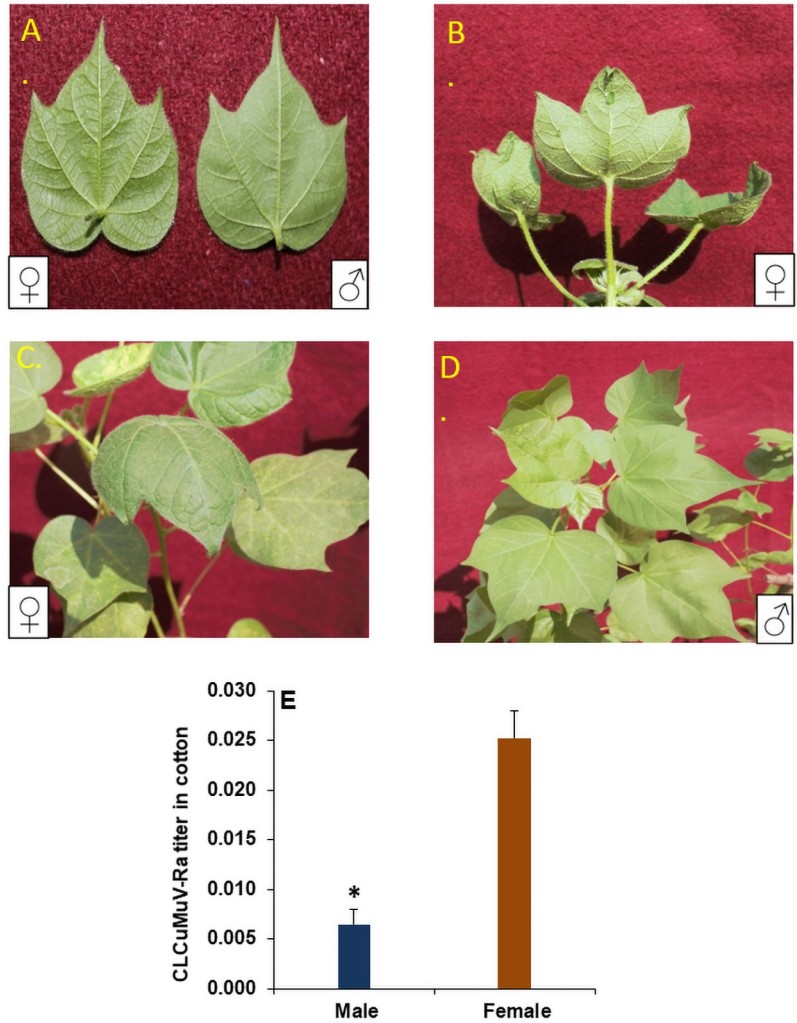

**Fig 2.** A. Early vein thickening in cotton plant after inoculation with viruliferous male and female whiteflies; B. Typical curling and enation in cotton after infection with viruliferous females; C. Downward curling in cotton plant after inoculating with viruliferous females; D. Symptomless or delayed symptoms in cotton plants after infection with viruliferous male; E. Relative amount of *Cotton leaf curl Multan virus*-Rajasthanin host plant normalized with *histone 3* as housekeeping gene for *G. hirsutum*. The error bars represent the standard deviation (n = 3) and *represents significant differences (P ≤ 0.05, Student's t-test).

inoculated plants showed only mild vein thickening even after one month of inoculation (Fig 2B–2D). The percent disease incidence (PDI) at the same point of time (14DPI) was also higher (37.5) in female inoculated plants compared to male inoculated plants. The appearance of symptoms in these plants was also supported by quantifying the relative expression of CLCuVcoat protein (CP) gene, which gave a qualitative estimate of relative CLCuV titer in plants inoculated by different *B. tabaci* sex. The viral load in the female inoculated cotton plants was significantly higher compared to males (Fig 2E) with 74.33% higher load with 20 released females compared to an equal number of males at 14 DPI. These results clearly indicate that both male and female whiteflies were able to acquire, retain and transmit CLCuV but quantitative variations in virus load among the sex could be one of the factors which make females efficient vectors compared to males. Despite equal virus acquisition time, the virus

load in males was significantly lower compared to females. This may be in support of the earlier studies suggesting higher acquisition and retention ability of females due to higher feeding capacity and larger size compared to male whiteflies [23]. Sex dependent interaction of begomoviruses and it whitefly vector has been reported in earlier studies [24–26] and few studies have also shown higher vector competence of females compared to male [27].

### Relative density of endosymbiont *Arsenophonus* Asia II-1 *B. tabaci* haplotype and its implications in virus transmission

Our first-hand results indicate that females are efficient virus transmitters due to the higher virus inoculum carried by it compared to males. However, we speculated some other possible factors which might be playing a key role in variable transmission efficiency of males and females based on our previous studies [21]. Our earlier studies implicate the role of endosymbiont *Arsenophonus* present in whitefly in virus transmission and thus infection status of this endosymbiont in individual sex might give some clues for sex-biased virus retention in *B. tabaci*. Relative densities of *Arsenophonus* inboth viruliferous and non- viruliferous males and females quantified using qRT-PCR through the expression of the 23SrRNA gene of the bacterium revealed significantly higher expression in females compared to males. The female non-viruliferous and viruliferous whiteflies seem to harbor significantly more *Arsenophonus* (55.9%) and (76.1%) than non-viruliferous and viruliferous male whitefly, respectively (Fig 3A). Further, feeding of sucrose diet supplemented with tetracycline (100 μg/ ml of diet) resulted in 76.8 and 85.1% elimination of *Arsenophonus* in male and female whiteflies compared to control, respectively (Fig 3B) In-depth studies are required to understand the reasons for sex-biased differential density of *Arsenophonus* in whitefly. Furthermore, the elimination of *Arsenophonus* from whitefly using tetracycline showed a 74.9% reduction in viral load in females compared to control whiteflies (Fig 3C). Similarly, in the case of *Arsenophonus* cured males 54.3% reduction in viral load was observed compared to control males (Fig 3C).

Facultative symbionts such as *Arsenophonus* and *Hamiltonella* are known to produce a GroEL chaperonin which protects the begomoviruses during its circulatory passage inside the hemolymph and plays a crucial role in vector-mediated virus transmission [7, 8]. The present study reports 58.3% higher expression of GroEL in viruleferous whiteflies compared to non-viruleferous irrespective of sex (Fig 4A). Further, the tetracycline mediated elimination of *Arsenophonus* also resulted in 76.7 and 68.4% downregulation in expression of GroEL in female and male whiteflies compared to control, respectively (Fig 4B and 4C). An earlier study with Asia-II-1 haplotype of *B. tabaci* also suggests that *Arsenophonus* produces GroEL protein which binds to the CLCuV particles inside the host and aids in the protective circulatory transmission of this virus to the host plant [13]. Moreover, female-biased facultative symbiont infections have been reported earlier in *Bemisia tabaci* and they tend to possess a higher load of a plant virus in comparison to males [28, 29]. Similarly, in the case of *Arsenophonus* cured males 54.3% reduction in viral load was observed compared to control males (Fig 3C). Our earlier study suggested that the Asia-II-1 haplotype of *B. tabaci* carries *Porteira* as primary endosymbiont and *Arsenophon*us and *Cardinium* as secondary endosymbionts, while this native culture lacks *Hamiltonella*, *Wolbachia*, *Rickettsia*, Those studies also revealed that the elimination of *Arsenophonus* in whitefly using tetracycline resulted in a reduction of CLCuV titer in *Arsenophonus—ve* whitefly, further the transmission of the virus in cotton plants was significantly reduced compared to *Arsenophonus +ve* whiteflies. Taking into account the role of *Arsenophonus* in virus transmission reported in earlier studies and the sex-biased density of this endosymbiont in whitefly, our studies indicate that the higher titer of virus in host female may be

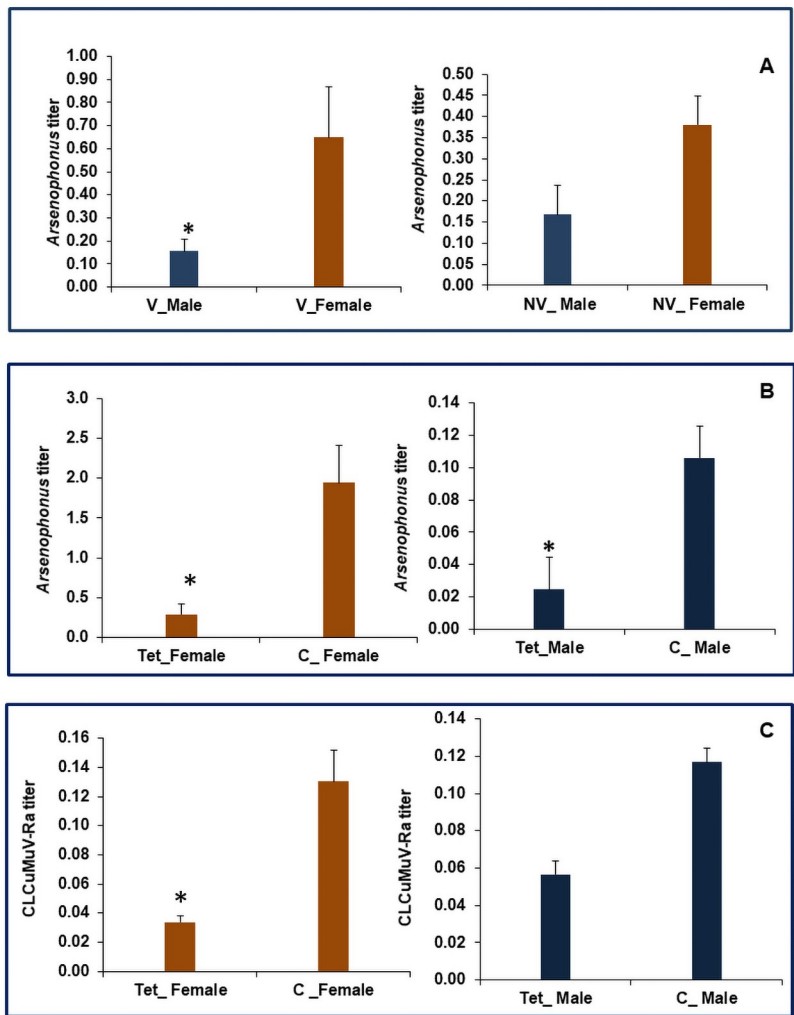

**Fig 3.** Relative amount of *Arsenophonus* in, A. viruliferous and non-viruliferous whitefly, B. Tetracycline fed females and males; C. Relative titer of *Cotton leaf curl Multan virus*-Rajasthanin tetracycline fed female and male whiteflies. The expression level was normalized using *B. tabaciβ-tubulin* as housekeeping gene. The error bars represent the standard deviation (n = 3) and *represents significant differences among male and female whiteflies (P ≤ 0.05, Student's t-test).

due to the higher retention of the virus with a higher level of GroEL protein produced compared to males.

## Sex-baised expression and functional validation of midgut proteins genes involved in virus transmission

Midgut protein genes such as *Cyclophilin*, *Knottin*, *Hsp40*, and *Hsp70* have been implicated in virus transmissionin earlier studies. The expression of these genes in non-viruliferous and viruliferous whiteflies revealed sex-biased expression across males and females. The relative expression of these genes seems to be associated with virus titer in host whitefly as revealed through the knockdown of respective genes through dsRNA feeding.

The expression of *Cyclophilin* in non-viruliferous male and females was statistically at par with each other (Fig 5A). However, the viruliferous female whitefly showed 57.75%

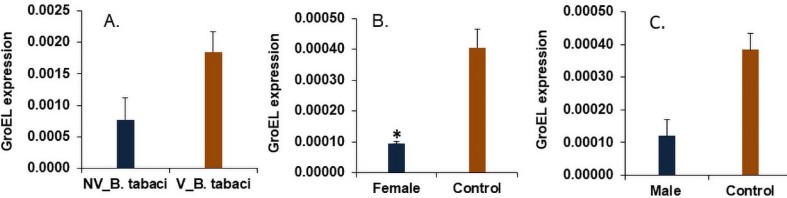

**Fig 4.** Relative expression of *GroEL* protein gene in A. Non-viruliferous and Viruliferous *Bemisia tabaci* B. Tetraccycline fed females C. Tetracycline fed males. The expression level was normalized with *β-tubulin* as housekeeping gene. The error bars represent the standard deviation (n = 3) and *represents significant differences among male and female whiteflies (P ≤ 0.05, Student's t-test).

significantly higher expression of *Cyclophilin* than the males (Fig 5A). The higher expression of this gene might be a consequence of the virus uptake in viruliferous whiteflies compared to non-viruliferous ones. In humans, *the Cyclophilin* gene plays a key role as modulators for virus replication [30]. It belongs to a protein family of peptidyl-prolyl cis-trans isomerase that plays a significant role in protein refolding, maturation, gene transcription, and cell signaling [31, 32]. The dsRNA mediated knockdown of *Cyclophilin* resulted in 73.2 and 65.7% reduction of target gene transcripts in female and male whiteflies compared to dsGFP fed control flies, respectively (Fig 5B). The dsRNA-mediated knockdown of *Cyclophilin* resulted in 48.3% and 19% decreased viral titer in females and male whiteflies (Fig 5C). This is indicative that *Cyclophilin* plays an active role in virus transmission through virus-vector interaction, which needs to be further investigated for in-depth understanding. Earlier studies have also demonstrated the role of *Cyclophilin* in virus transmission as the knockdown *Cyclophilin* in *B. tabaci* resulted in a 43% decrease of TYLCV transmission [14]. Our previous study also reported that the plants exposed to *Cyclophilin* silenced whiteflies contained lower numbers of viral transcripts compared to control [21]. Thus the previous studies establish a direct relationship of *Cyclophilin* with virus transmission in different organisms including insects, and the sex- biased differential expression further supports variable transmission efficiency among transmission by male and female whiteflies.

The present studies reveal the higher expression of *Knottin* in males compared to females in both viruliferous and non-viruliferous whitefly with non-viruliferous males exhibiting 64.84% higher expression compared to females. Similarly, viruliferous males showed 31.19% higher expressions of *Knottin* compared to female whitefly (Fig 6A). Feeding of dsKnottin resulted in 84.5 and 79.8% significant decrease in mRNA levels of *Knottin* in female and male whiteflies compared to dsGFP fed control flies, respectively (Fig 6B). *Knottins* are a structural family of proteins that contain small proteins rich in disulphide with a knotted appearance [33]. Previous studies demonstrate that *Knottins* are involved in maintaining the amount of virus associated with whitefly vectors to a level that may not be deleterious to the host [16]. Silencing of *Knottin* resulted in a significant increase in viral expression in viruliferous female whiteflies, while a similar but slight increase of viral load was also observed in viruliferous male whiteflies. Downregulation of K*nottin* resulted in 80.7% more viral load in females when compared to GFP-fed female whiteflies (Fig 6C). However, knockdown of *Knottin* in males showed 20.7% more viral transcripts when compared to control male flies (Fig 6C). The earlier studies have also reported an increase in virus ingestion by the whitefly or transmission efficiency by several-fold through silencing *Knottin* in whitefly by feeding them on dsRNA [16]. Our previous studies also suggest that dsRNA-mediated knockdown of *Knottin* in whiteflies resulted in 1.9 fold increase in virus titer compared to control and ultimately resulted in 2.0 fold higher transmission efficiency of virus in cotton plants by the dsRNA mediated *Knottin* knockdown

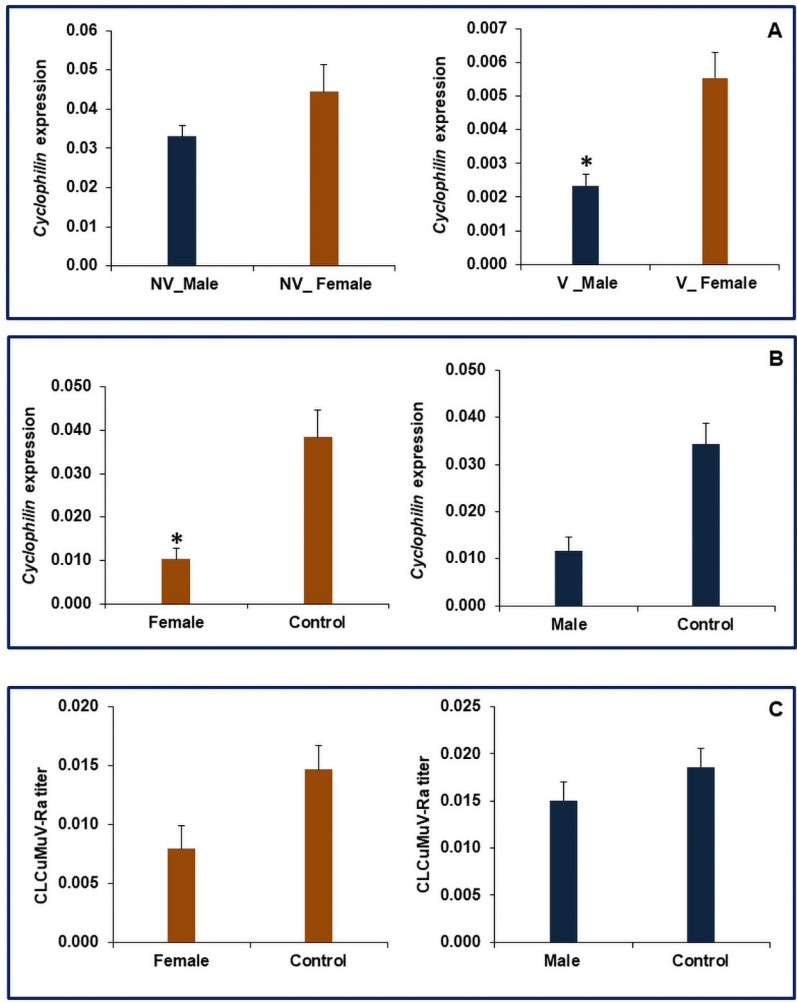

**Fig 5.** Relative expression of *Cyclophilin in B. tabaci*, A. Non-viruliferous and Viruliferous whiteflies, B. dsRNA fed female and male whiteflies; C. Relative titer of *Cotton leaf curl Multan virus*-Rajasthan in *Cyclophilin* knockdown viruliferous female and male whiteflies. The expression level was normalized with *β-tubulin* as housekeeping gene. Error bars represent the standard deviation (n = 3) and *represents significant differences in expression (P ≤ 0.05, Student's t-test).

whiteflies [21]. Thus the lower expression of *Knottin* in females may be another possible factor responsible for higher virus transmission efficiency compared to males.

In a similar context present study reveals that the relative expression of *Hsp's* was higher in males as compared to females either in non-viruliferous or viruliferous whiteflies. The relative expression of *Hsp40* was 82.05 and 61.61% in non-viruliferous and viruliferous males, respectively, which was significantly higher compared to females (Fig 7A). Similarly, in the case of *Hsp70*, the non-viruliferous and viruliferous males showed 55.19% and 58.28%, respectively higher expression than females (Fig 8A). The feeding of dsRNA against *Hsp 40* resulted in 80.7 and 86.9% reduction in mRNA levels of this gene in female and male flies compared to control, respectively (Fig 7B). The corresponding figures for *Hsp70* were 79.2 and 82.3% in female and male whiteflies compared to dsGFP fed control flies, respectively (Fig 8B). Heat shock proteins are universally present in most of the organisms and have been nomenclature based on their molecular weight into *Hsp100*, *Hsp90*, *Hsp70*, *Hsp60*, *Hsp40*, and small *Hsps* (size>30kDa),

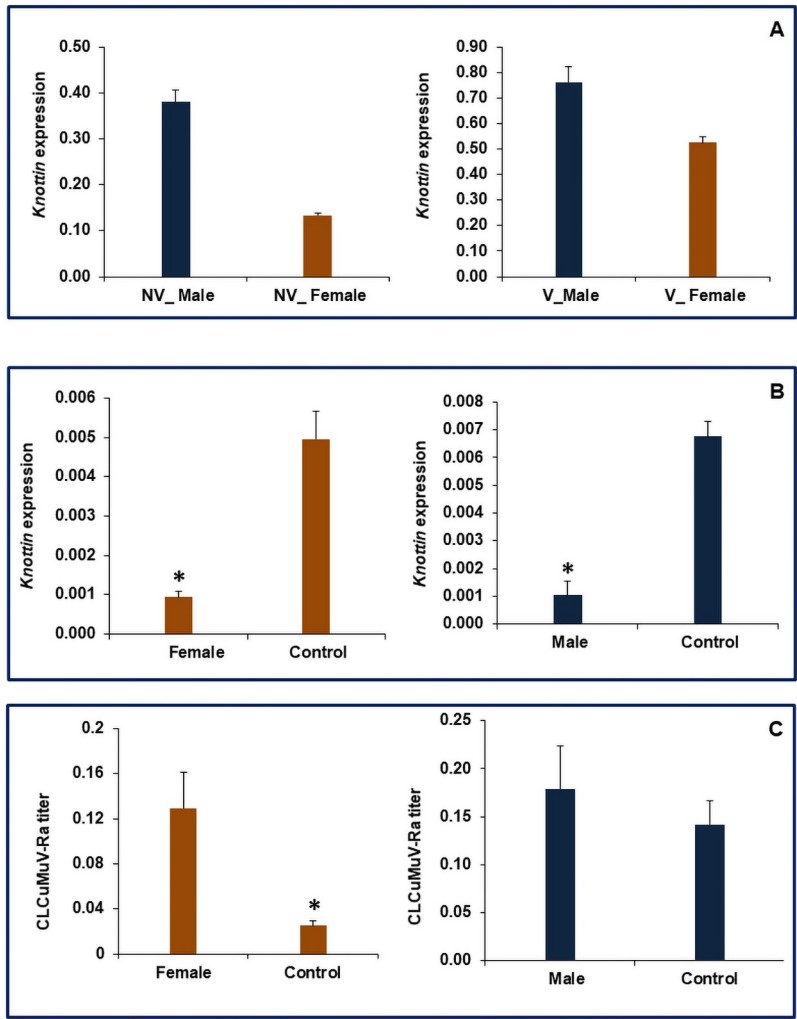

**Fig 6.** Relative expression of *Knottin* in *B. tabaci*, A. Non-viruliferous and Viruliferous whiteflies, B. dsRNA fed female and male whiteflies; C. Relative titerof *Cotton leaf curl Multan virus*-Rajasthanin *Knottin* knockdown viruliferous female and male whiteflies. The expression level was normalized with *β-tubulin* as housekeeping gene. Error bars represent the standard deviation (n = 3) and *represents significant differences in expression (P ≤ 0.05, Student's t-test).

which is represented by the numeric prefix in the name of respective *Hsp* [34]. *Hsp70* isoforms have also been reported to play a role in the entry of dengue virus, its replication, and biogenesis in mosquitoes [35]. Our previous studies suggest that silencing of *Hsp40* and *Hsp70* in whitefly affected the virus titer in whitefly and consequently increased transmission of CLCu-MuV-Rain cotton [21]. The activity of both *Hsp40* and *Hsp70* is interdependent as *Hsp40* functions as co-chaperon acting as a DnaJ dimer which stimulates the ATPase activity of *Hsp70* [36, 37]. It is thus inferred from these studies that the higher expression of *Hsp40* and 70 might be contributing to poor transmission efficiency of CLCuMuV-Raby males compared to female whiteflies. Further, the dsRNA-mediated silencing of *Hsp70* and *Hsp40* resulted in an increase in viral transcripts in viruliferous females as compared to males. Knockdown of *Hsp 70* and *Hsp 40* in females resulted in 89.8 and 74%, respectively more viral load compared to GFP-fed female whiteflies (Figs 7C and 8C). A similar increase of 22 and 18% were observed with

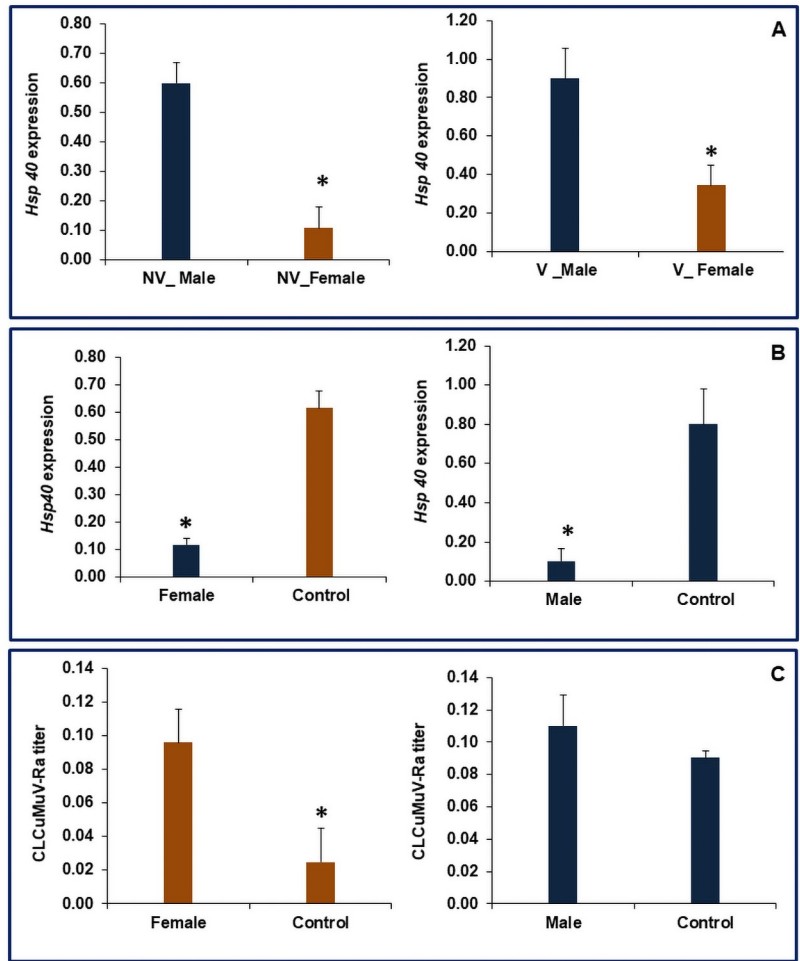

**Fig 7.** Relative expression of *Hsp40* in *B. tabaci*, A. Non-viruliferous and viruliferous whiteflies, B. dsRNA fed female and male whiteflies; C. Relative titer of *Cotton leaf curl Multan virus*-Rajasthanin *Hsp40* knockdown viruliferous female and male whiteflies. The expression level was normalized with *β-tubulin* as housekeeping gene. Error bars represent the standard deviation (n = 3) and *represents significant differences in expression (P ≤ 0.05, Student's t-test).

dsRNA mediated knockdown of *Hsp70 and Hsp40*, respectively in males when compared to control. Previous studies reported that *Hsp70 and Hsp40* act by minimizing the potential long-term harmful effects of the virus on the whitefly [9]. Our previous studies with bulk whitefly populations confirm and support our recent results in which silencing the expression of *Hsp70* and *Hsp 40* resulted in an increase in CLCuMuV-Ratiter and transmission by vector *B. tabaci* [21]. Sex-biased expression of *Hsp70* and *Hsp40* and previous studies suggest that these heat shock proteins may be acting as a negative regulator of the virus which inhibits the circulatory passage of virus within the host [15, 21].

The present study first time reports the clues in sex-biased transmission efficiency of CLCuMuV-Rain a female and male Asia-II-1 haplotype of *B. tabaci*. Based on the results it may be inferred that there are multiple factors that play a role towards variable efficiency of the female towards the transmission of the virus. The large size of female whitefly and consequently more feeding compared to males may be attributed to more virus accumulation compared to males. For successful transmission, the virus has to pass certain barriers such as the midgut wall to

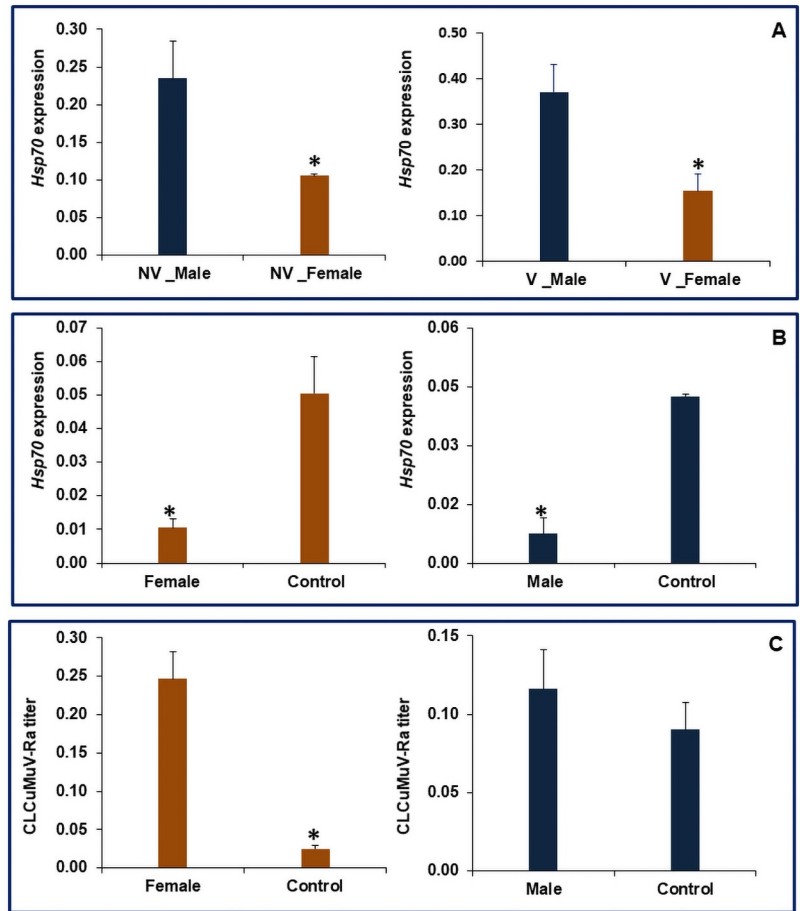

**Fig 8.** Relative expression of *Hsp70* in *B. tabaci*, A. Non-viruliferous and Viruliferous whiteflies, B. dsRNA fed female and male whiteflies; C. Relative titer of *Cotton leaf curl Multan virus*-Rajasthanin *Hsp70* knockdown viruliferous female and male whiteflies. The expression level was normalized with *β-tubulin* as housekeeping gene. Error bars represent the standard deviation (n = 3) and *represents significant differences in expression (P ≤ 0.05, Student's t-test).

reach into the hemolymph and finally to accumulate in the primary salivary gland from where it reaches into the host plant [38, 39]. Therefore variable viral movement and retention ability while following a circulatory route as perceived from viral load within the male and female whiteflies may contribute to differential transmission patterns. The more viral load retained by female whiteflies due to multiple factors studied might be leading to the passage of more viruses from hemolymph to primary salivary glands compared to males. Variance in vector competence has been speculated to impart differential capacity of the virus to cross barriers within the body of different whitefly species [38, 39]. The persistent transmission of the virus within the whitefly is also mediated by the endosymbiont partner which provides a protective barrier to the virus while its passage from hemolymph [8, 40]. Higher density or more accumulation of endosymbiont *Arsenophonus* in females may be a contributory factor towards higher transmission efficiency of females. The low expression of *Cyclophilin* and higher expression of *Hsp40*, *Hsp70* and *Knottin* in males might be another factor for the low transmission efficiency of male whiteflies. Taken together, we found that in the Asia II-1 haplotype of whiteflies, midgut proteins and facultative endosymbiont *Arsenophonus* has a profound influence in

shaping sex-biased differential virus transmission (S1 Graphical abstract). However, variable endosymbiont density or variable expression mid-gut protein genes across sex is a researchable issue and needs in-depth studies to better understand virus-vector sex interactions.

## Supporting information

**S1 Fig.** A. Maintenance of virus free plants inside insect proof cages B. Clip inoculation setup for whitefly inoculation C. Cup cages for whitefly inoculation.
(PDF)

**S2 Fig. Morphological distinction of male (♂) and female (♀) whiteflies based upon size and shape of ovipositor.**
(PDF)

**S3 Fig. Artificial feeding set up for oral feeding access to whiteflies.**
(PDF)

**S1 Graphical abstract.**
(TIF)

## Acknowledgments

We thank Punjab Agricultural University, Ludhiana and Ministry of Science and Technology, New Delhi for the support to conduct this study.

## Author Contributions

**Conceptualization:** Satnam Singh.

**Formal analysis:** Satnam Singh.

**Investigation:** Ikbalpreet Singh, Ramandeep Kaur.

**Methodology:** Ikbalpreet Singh, Ramandeep Kaur.

**Resources:** Ashok Kumar.

**Supervision:** Satnam Singh, Abhishek Sharma.

**Writing – original draft:** Ikbalpreet Singh, Ramandeep Kaur, Satnam Singh, Abhishek Sharma.

**Writing – review & editing:** Ashok Kumar, Satnam Singh.

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
