## [Decision Letter · Decision Letter 0]

22 Jun 2021

PONE-D-21-14850

Is Cotton Leaf Curl Virus transmission by Bemisia tabaci (Gennadius) (Hemiptera: Aleyrodidae) sex-biased?

PLOS ONE

Dear Dr. Singh,

Thank you for submitting your manuscript to PLOS ONE. After careful consideration, we feel that it has merit but does not fully meet PLOS ONE’s publication criteria as it currently stands. Therefore, we invite you to submit a revised version of the manuscript that addresses the points raised during the review process.

We look forward to receiving your revised manuscript.

Kind regards,

Rajarshi Gaur

Academic Editor

PLOS ONE

Journal Requirements:

2. We note that you have referenced (ie. Bewick et al. [5]) which has currently not yet been accepted for publication. Please remove this from your References and amend this to state in the body of your manuscript: (ie “Bewick et al. [Unpublished]”) as detailed online in our guide for authors

3. We note that Figure 2 and Supplementary Figures 1, 2 and 3 in your submission contain copyrighted images. All PLOS content is published under the Creative Commons Attribution License (CC BY 4.0), which means that the manuscript, images, and Supporting Information files will be freely available online, and any third party is permitted to access, download, copy, distribute, and use these materials in any way, even commercially, with proper attribution. For more information, see our copyright guidelines: http://journals.plos.org/plosone/s/licenses-and-copyright.

1. You may seek permission from the original copyright holder of Figures 2 and Supplementary Figures 1, 2 and to publish the content specifically under the CC BY 4.0 license. 

Reviewers' comments:

Reviewer's Responses to Questions

**Comments to the Author**

1. Is the manuscript technically sound, and do the data support the conclusions?

Reviewer #1: Partly

Reviewer #2: Partly

2. Has the statistical analysis been performed appropriately and rigorously? 

Reviewer #1: Yes

Reviewer #2: Yes

3. Have the authors made all data underlying the findings in their manuscript fully available?

Reviewer #1: Yes

Reviewer #2: Yes

4. Is the manuscript presented in an intelligible fashion and written in standard English?

Reviewer #1: Yes

Reviewer #2: Yes

5. Review Comments to the Author

Reviewer #1: Review Comments to the Authors:

This study by Singh et al focuses on the sex-biased transmission efficiency of Cotton Leaf Curl Virus (CLCuV) by Bemisia tabaci. The gender dependent differential variation was validated by using quantitative real-time PCR through the relative expression of CLCuV coat protein. The female whiteflies acquired and transmitted higher viral load compared to males. The authors identified that the higher density of Arsenophonus and differential expression of Cyclophilin, Knottin, Hsp40, Hsp70 of the midgut genes may possibly contribute to higher vector competency to females compared to males. This study advances our understanding to key players involved in sex biased CLCuV transmission. Overall, the study is novel and well conceptualized, however there are some points which need to be addressed before consideration to publish.

Major comments:

1. The CLCuV coat protein real-time PCR primers used for this study should be tested on non-infected whitefly as well as non-infected plant to ensure there is not any non-specific binding.

2. Line 264-266 explained Fig 3c and indicating expression levels of CLCuV coat protein, but the figure legends are indicating these are Arsenophonus levels. Please update the figure legends.

3. As reported, there is differential level of expression of the genes in male and female so the efficiency of knock down (KD) using same amounts dsRNA for both male as well as female could be different and that may affect the observations. Indeed, the transmission efficiency of KD female is reduced more compared to KD males which is in consistent with the study. Ideally, it is important to normalize the gene expression in knockdown male as well as females. In short, authors should provide data for the expression levels of Arsenophonus in tetracycline fed/control male as well as female to validate the knock down for Arsenophonus. Also, it would have been interesting to have the comparative expression levels of GroEL during acquisition or transmission to strengthen and complement the data.

4. The authors should provide data for the expression levels of Cyclophilin in dsRNA fed/control male and female to validate the authenticity of knockdowns. Similar data should be provided for Knottin, Hsp40, Hsp70 as proof of concept.

5. Fig.6c data plot should be in comparison to male versus GFP fed control. Similarly, for fig.6d data plot should be comparison to female versus GFP fed control. Please update the labels on data plots.

6. It was quite evident that the expression levels of Arsenophonus, Knottin, Hsp40, Hsp70 in viruliferous flies are higher in comparison to non- viruliferous flies irrespective of gender. But the expression levels of Cyclophilin are lower in viruliferous flies in comparison to non-viruliferous flies. The authors should address that in the manuscript.

Minor comments:

1. The male and female should be represented with different color for better representation of the graphs.

2. The authors should label Y-axis for all the data plots for consistency throughout the manuscript.

3. The statistics analysis is missing for the graphs Fig.1 c, Fig.3 d, Fig.4 a, c, d, Fig.5 a, b, c, Fig.6 d, Fig.7 c.

4. Primers in Table 1 should be cross validated as the forward and reverse primers for HSP40 cannot be identical.

5. Please provide the details (accession number or Uniprot ID) in methods section for the CLCuV coat protein sequence used for designing the primers for real-time PCR. Also, adding details of the source of Cotton Leaf Curl Rajasthan Virus would be better.

6. It is also recommended to provide additional details in methods sections for the software or tools used to analyze real time data, statistical and image analysis.

7. Line 220 please provide the full form for DOI.

Reviewer #2: Whitefly, Bemisia tabaci (Gennadius) is an important pest of cotton as vector of Cotton Leaf Curl Virus (CLCuV) and the paper showed B. tabaci AsiaII-1 haplotype showing higher virus transmission efficiency of females than males. The transmission is related to the density of endosymbiont Arsenophonus, but the expression of some midgut proteins genes i.e. Cyclophilin, Knottin, Hsp40, Hsp70 may be possibly associated with the vector competency of both sex. All the results might not give us explicit conclusions that females treated by knocked-down or antibiotics could directly show higher competency to transmit the virus.

My suggestion is that authors just focus on the role of one of endosymbionts and its midgut protein on the sex-biased transmission of whitefly. However, the title “Is Cotton Leaf Curl Virus transmission by Bemisia tabaci (Gennadius) (Hemiptera: Aleyrodidae) sex-biased?” does not fully explain and cover all main ideas of the manuscript, i.e. the relation the sex-biased of transmission with endosymbionts and their proteins.

6. PLOS authors have the option to publish the peer review history of their article (what does this mean?). If published, this will include your full peer review and any attached files.

Reviewer #1: No

Reviewer #2: No

---

## [Author Response · Author response to Decision Letter 0]

28 Jul 2021

Editor’s comments:

1 Please ensure that your manuscript meets PLOS ONE's style requirements, including those for file naming

Manuscript has been modified according to Plos one style as suggested.

2 We note that you have referenced (ie. Bewick et al. [5]) which has currently not yet been accepted for publication. Please remove this from your References and amend this to state in the body of your manuscript: (ie “Bewick et al. [Unpublished]”) as detailed online in our guide for authors

Reference has been deleted as per suggestion 

3. We note that Figure 2 and Supplementary Figures 1, 2 and 3 in your submission contain copyrighted images. All PLOS content is published under the Creative Commons Attribution License (CC BY 4.0), which means that the manuscript, images, and Supporting Information files will be freely available online, and any third party is permitted to access, download, copy, distribute, and use these materials in any way, even commercially, with proper attribution.

Figure 2 and Supplementary Figures 1, 2 and 3 have been clicked by us in our lab/ greenhouse setup. So we do not need to seek any permissions for using these figures in our manuscript.

However, S1 Fig. contains pictures of cages purchased from Bugdrom Store (Taiwan). The permission from the manufacturer has been sought to use the picture of cages in manuscript (permission under the CC BY 4.0 license has been attached)

Reviewer 1:

1. The CLCuV coat protein real-time PCR primers used for this study should be tested on non-infected whitefly as well as non-infected plant to ensure there is not any non-specific binding.

The CLCuV coat protein primers has been validated on non-viruliferous and viruliferous whiteflies and plants. Details added in Fig 1.

2. Line 264-266 explained Fig 3c and indicating expression levels of CLCuV coat protein, but the figure legends are indicating these are Arsenophonus levels. Please update the figure legends.

Done

3As reported, there is differential level of expression of the genes in male and female so the efficiency of knock down (KD) using same amounts dsRNA for both male as well as female could be different and that may affect the observations. Indeed, the transmission efficiency of KD female is reduced more compared to KD males which are in consistent with the study. Ideally, it is important to normalize the gene expression in knockdown male as well as females. In short, authors should provide data for the expression levels of Arsenophonus in tetracycline fed/control male as well as female to validate the knock down for Arsenophonus. Also, it would have been interesting to have the comparative expression levels of GroEL during acquisition or transmission to strengthen and complement the data.

We thanks reviewer for this comment. The knocked down after tetracycline feeding in female and male relative to control female and male whiteflies has been validated and added in main text along with figure and statistical data. As per suggestion comparative expression levels of GroEL during acquisition or transmission has been conducted and incorporated in Fig 4. 

4. The authors should provide data for the expression levels of Cyclophilin in dsRNA fed/control male and female to validate the authenticity of knockdowns. Similar data should be provided for Knottin, Hsp40, Hsp70 as proof of concept

The relative expression Cyclophilin, Knottin, Hsp40 and Hsp70 in dsRNA fed and control male and female has been validated using RT-PCR and statistical analysis was performed by Student t-Test. The data has been added in main text body and Fig 5, 6, 7 and 8

5. Fig.6c data plot should be in comparison to male versus GFP fed control. Similarly, for fig.6d data plot should be comparison to female versus GFP fed control. Please update the labels on data plots.

Fig. 6c and 6d has been rephrased to updated to panel Fig. 7C and labels have been updated as per suggestions.

Minor Comments

1. The male and female should be represented with different color for better representation of the graphs.

Male and female has been represented by different colours in all the figures 

2. The authors should label Y-axis for all the data plots for consistency throughout the manuscript.

Done 

3. The statistics analysis is missing for the graphs Fig.1 c, Fig.3 d, Fig.4 a, c, d, Fig.5 a, b, c, Fig.6 d, Fig.7 c.

The data have been analysed by Student’s T test and this has been mentioned in the text

4. Primers in Table 1 should be cross validated as the forward and reverse primers for HSP40 cannot be identical.

This has been changed as suggested.

5. Please provide the details (accession number or Uniprot ID) in methods section for the CLCuV coat protein sequence used for designing the primers for real-time PCR. Also, adding details of the source of Cotton Leaf Curl Rajasthan Virus would be better.

Accessions have been mentioned in the text. Inadvertently we have written the virus name as Cotton Leaf Curl Rajasthan Virus in place of Cotton Leaf Curl Multan-Rajasthan Virus and this is the correct classification as per ICTV Code. The virus strain has been continuously maintained on cotton plants and is crosschecked time to time for its purity.

6. It is also recommended to provide additional details in methods sections for the software or tools used to analyze real time data, statistical and image analysis.

Details have been added in materials methods.

7. Line 220 please provide the full form for DOI.

The DOI has been replaced by DPI (Days Post Inoculation)

Reviewer 2

1. Whitefly, Bemisia tabaci (Gennadius) is an important pest of cotton as vector of Cotton Leaf Curl Virus (CLCuV) and the paper showed B. tabaci AsiaII-1 haplotype showing higher virus transmission efficiency of females than males. The transmission is related to the density of endosymbiont Arsenophonus, but the expression of some midgut proteins genes i.e. Cyclophilin, Knottin, Hsp40, Hsp70 may be possibly associated with the vector competency of both sex. All the results might not give us explicit conclusions that females treated by knocked-down or antibiotics could directly show higher competency to transmit the virus. My suggestion is that authors just focus on the role of one of endosymbionts and its midgut protein on the sex-biased transmission of whitefly. However, the title “Is Cotton Leaf Curl Virus transmission by Bemisia tabaci (Gennadius) (Hemiptera: Aleyrodidae) sex-biased?” does not fully explain and cover all main ideas of the manuscript, i.e. the relation the sex-biased of transmission with endosymbionts and their proteins.

This is a good suggestion and keeping in view the reviewers comments we have revised the title of manuscript title highlighting Arsenophonus and gut proteins.

---

## [Decision Letter · Decision Letter 1]

4 Oct 2021

PONE-D-21-14850R1Differential expression of gut protein genes and population density of Arsenophonus contributes to sex-biased transmission of Bemisia tabaci vectored Cotton leaf curl virusPLOS ONE

Dear Dr. Singh,

Thank you for submitting your manuscript to PLOS ONE. After careful consideration, we feel that it has merit but does not fully meet PLOS ONE’s publication criteria as it currently stands. Therefore, we invite you to submit a revised version of the manuscript that addresses the points raised during the review process.

We look forward to receiving your revised manuscript.

Kind regards,

Rajarshi Gaur

Academic Editor

PLOS ONE

Journal Requirements:

Reviewers' comments:

Reviewer's Responses to Questions

**Comments to the Author**

1. If the authors have adequately addressed your comments raised in a previous round of review and you feel that this manuscript is now acceptable for publication, you may indicate that here to bypass the “Comments to the Author” section, enter your conflict of interest statement in the “Confidential to Editor” section, and submit your "Accept" recommendation.

Reviewer #1: (No Response)

Reviewer #3: All comments have been addressed

2. Is the manuscript technically sound, and do the data support the conclusions?

Reviewer #1: Yes

Reviewer #3: Yes

3. Has the statistical analysis been performed appropriately and rigorously? 

Reviewer #1: Yes

Reviewer #3: N/A

4. Have the authors made all data underlying the findings in their manuscript fully available?

Reviewer #1: Yes

Reviewer #3: Yes

5. Is the manuscript presented in an intelligible fashion and written in standard English?

Reviewer #1: Yes

Reviewer #3: Yes

6. Review Comments to the Author

Reviewer #1: This study by Singh et al, provides substantial evidence of correlation between sex based differential gene expression and transmission efficiency of Cotton Leaf Curl Virus (CLCuV) by Bemisia tabaci. The study was conducted using B. tabaci Asia II-1 halotype and the virus transmission efficiency of females was reported significantly higher over males. The authors have addressed majority of the comments from both the reviewers. However, I have mentioned few important concerns which should be addressed before consideration for publishing. Therefore, I would recommend acceptance with minor revision.

1- The color codes for the data plots are not consistent for example female are orange and controls are white in Fig-1 but in Fig 3B and 3C both male and female are blue, and controls became orange. Please try to maintain consistent colors throughout the manuscript.

2- Please provide the accession number if available for the coat protein sequence of the CLCuMuV-Ra which was used to design the real time primers for this study.

3- The real time PCR reaction conditions mentioned in lines 119-121 are different from lines 150-152, but both were used for the detection of coat protein either from whiteflies or from infected plants. Please justify.

4- Lines 229 and 230 please correct nl to microliters.

5- Fig1 E the fluid loss assay data is not clear as blue dots are not easy to visualize.

6- Lines 279-280 belong to Fig 3 legends to my understanding. Please correct accordingly.

7- The relative levels of Arsenophonus in non-viruliferous whiteflies are significantly higher than viruliferous whiteflies (Fig 3A) which is not corelating with the relative levels of GroEL. The relative amounts of GroEL are lower in non-viruliferous whiteflies in comparison with viruliferous whiteflies (Fig 4A) Please cross-check if there is labelling error in data plots of Fig3A or justify.

8- Line 343 the expression levels of cyclophilin in non-viruliferous males and females are not at par in Fig 5A. As Fig 5A data plots clearly indicate that females have higher levels of cyclophilin in viruliferous as well as non-viruliferous whiteflies. Please modify accordingly.

9- Fig 7B the control females have higher levels of Hsp40 than control males while Fig 7A clearly states females express lesser amounts of Hsp40. Similarly, Fig 8B the control females have slightly higher levels of Hsp70 than control males which is contradictory to data in 8A. Please correct the controls accordingly to support the data for 7A and 8A.

10- Fig 5C, 6C, 7C, 8C the control females have lower virus transmission rate than control males while Fig 1 clearly indicates female have higher transmission rate than males it is not consistent with the hypothesis throughout the manuscript. Please add corrections accordingly or provide justification.

Reviewer #3: The MS entitled "Differential expression of gut protein genes and population density of Arsenophonus contributes to sex-biased transmission of Bemisia tabaci vectored Cotton leaf curl virus" is in present form is suitable for the publication

7. PLOS authors have the option to publish the peer review history of their article (what does this mean?). If published, this will include your full peer review and any attached files.

Reviewer #1: No

Reviewer #3: No

---

## [Author Response · Author response to Decision Letter 1]

16 Oct 2021

We are grateful to the reviewers for spending their time for the critical and valuable comments in the manuscript. We extend our heartfelt thanks to Reviewer 1 for critical analysis of the figures which has helped us to correct the inadvertent cut and copy mistakes in the Fig 3A and 7 B. We have tried to address each and every query raised by the reviewer. The point to point rebuttal has been given below:

Reviewer comment:The color codes for the data plots are not consistent for example female are orange and controls are white in Fig-1 but in Fig 3B and 3C both male and female are blue, and controls became orange. Please try to maintain consistent colors throughout the manuscript

Authors response: Colour codes of the figures have been modified as suggested. 

Reviewer Comment: Please provide the accession number if available for the coat protein sequence of the CLCuMuV-Ra which was used to design the real time primers for this study.

Authors response: The accession no of CLCuMuV-Ra virus is provided in line no 98.As per suggestion this is also incorporated in lines 131and 132

Reviewer Comment: The real time PCR reaction conditions mentioned in lines 119-121 are different from lines 150-152, but both were used for the detection of coat protein either from whiteflies or from infected plants. Please justify.

Authors Response: The conditions have been made uniform. Apologies for this error, inadvertently the protocol for three step amplification, which is also commonly used in our lab was written at 119-121. 

Reviewer Comment: Lines 229 and 230 please correct nl to microliters

Authors Response: The total diet consumption in male and female (n=50) whiteflies was 2.4 and 3.4 µl after subtracting the diet loss from control (Fig 1E) and the diet consumption was converted in nl/ adult. So the nl is the correct representation. The same has been explained in text section.

Reviewer Comment: Fig1 E the fluid loss assay data is not clear as blue dots are not easy to visualize

Authors Response: The images have been converted into tiff format and are clearer. Thanks to reviewer for this comments as we have missed to put the methodology for this experiment and the same has been incorporated in the manuscript under “Feeding and Fluid loss assay”

Reviewer Comment: Lines 279-280 belong to Fig 3 legends to my understanding. Please correct accordingly

Authors Response: The subtitle of the next paragraph was merged with Fig. 2 legends. It has been corrected by spacing.

Reviewer comment: The relative levels of Arsenophonus in non-viruliferous whiteflies are significantly higher than viruliferous whiteflies (Fig 3A) which is not corelating with the relative levels of GroEL. The relative amounts of GroEL are lower in non-viruliferous whiteflies in comparison with viruliferous whiteflies (Fig 4A) Please cross-check if there is labelling error in data plots of Fig3A or justify.

Authors Response: We appreciate the reviewer’s critical analysis for Fig 3 A and & 7 B as both figures were wrongly pasted from excel. In Fig 3A the cq-values were mistakenly plotted which shows huge variability across the y-axis of plots. The graphs for Fig 3A has been replotted with correct values and incorporated in the Manuscript. 

Reviewer comment: Line 343 the expression levels of cyclophilin in non-viruliferous males and females are not at par in Fig 5A. As Fig 5A data plots clearly indicate that females have higher levels of cyclophilin in viruliferous as well as non-viruliferous whiteflies. Please modify accordingly.

Authors Response: Here we mean to say that the expression of cyclophilin in non-viruliferous males and females are statistically at par with each other as we these values are not significantly different as per Student’s t test. We have reformatted the line by addition ‘statistically at par with each other’

Reviewer Comment: Fig 7B the control females have higher levels of Hsp 40 than control males while Fig 7A clearly states females express lesser amounts of Hsp40

Authors Response: As mentioned earlier, the inadvertent mistake in plotting Fig. 7B, the data has been crosses checked and the correct plot has been incorporated in the manuscript. The percent knockdown 80.7 was correctly mentioned in the text earlier. 

Reviewer Comment: Similarly, Fig 8B the control females have slightly higher levels of Hsp70 than control males which is contradictory to data in 8A. Please correct the controls accordingly to support the data for 7A and 8A. 

Fig 5C, 6C, 7C, 8C the control females have lower virus transmission rate than control males while Fig 1 clearly indicates female have higher transmission rate than males it is not consistent with the hypothesis throughout the manuscript. Please add corrections accordingly or provide justification.

Authors Response: 

We agree to the reviewer perspective but each graph of panel is as a result of one experimentation under one set of conditions. It is not possible to compare the y-axis of the every plot in each panel of the figures or even across the figures. The reason is that every graph is a result of different biological sample under different set of experimental conditions which gives variation in housekeeping gene and PCR efficiency in spite of all efforts to keep the things same throughout the experimentation process. This can also be visualized from the variation in Standard deviation across all the figures. The results were interpreted on the basis of target gene in male and female relative to its reference control normalized with housekeeping gene at that point of time or that very single qPCR reaction so the two figures in each panel represent a relative expression in two different set of qPCR reactions.

Moreover, for Fig 8B slighter variation is clearly indicated by the ±0.01 value of SD bar in female control as compared to male controls.Thus we again emphasis that these graphs are generated on the basis of multiple experiments and replicates, the comparison of Y-axis values across graphs, which show slight experimental variations in Fig 5C, 6C, 7C, 8C is not possible.

---

## [Editor Report · Decision Letter 2]

19 Oct 2021

Differential expression of gut protein genes and population density of Arsenophonus contributes to sex-biased transmission of Bemisia tabaci vectored Cotton leaf curl virus

PONE-D-21-14850R2

Dear Dr. Singh,

We’re pleased to inform you that your manuscript has been judged scientifically suitable for publication and will be formally accepted for publication once it meets all outstanding technical requirements.

Kind regards,

Rajarshi Gaur

Academic Editor

PLOS ONE

---

## [Editor Report · Acceptance letter]

26 Oct 2021

PONE-D-21-14850R2 

Differential expression of gut protein genes and population density of*Arsenophonus* contributes to sex-biased transmission of *Bemisia tabaci*vectored *Cotton leaf curl virus*

Dear Dr. Singh:

I'm pleased to inform you that your manuscript has been deemed suitable for publication in PLOS ONE. Congratulations! Your manuscript is now with our production department. 

Kind regards, 

on behalf of

Professor Rajarshi Gaur 

Academic Editor

PLOS ONE